# Investigating the Viability of Epithelial Cells on Polymer Based Thin-Films

**DOI:** 10.3390/polym13142311

**Published:** 2021-07-14

**Authors:** Boštjan Vihar, Jan Rožanc, Boštjan Krajnc, Lidija Gradišnik, Marko Milojević, Laura Činč Ćurić, Uroš Maver

**Affiliations:** 1Faculty of Medicine, Institute of Biomedical Sciences, University of Maribor, Taborska Ulica 8, SI-2000 Maribor, Slovenia; jan.rozanc@um.si (J.R.); bostjan.krajnc@um.si (B.K.); lidija.gradisnik@um.si (L.G.); marko.milojevic1@um.si (M.M.); laura.cinc@um.si (L.Č.Ć.); uros.maver@um.si (U.M.); 2IRNAS d.o.o, Limbuška Cesta 76b, SI-2000 Maribor, Slovenia; 3Department of Pharmacology, Faculty of Medicine, University of Maribor, Taborska Ulica 8, SI-2000 Maribor, Slovenia

**Keywords:** polymers, thin films, viability, morphology, HUVEC, HUIEC

## Abstract

The development of novel polymer-based materials opens up possibilities for several novel applications, such as advanced wound dressings, bioinks for 3D biofabrication, drug delivery systems, etc. The aim of this study was to evaluate the viability of vascular and intestinal epithelial cells on different polymers as a selection procedure for more advanced cell-polymer applications. In addition, possible correlations between increased cell viability and material properties were investigated. Twelve polymers were selected, and thin films were prepared by dissolution and spin coating on silicon wafers. The prepared thin films were structurally characterized by Fourier transform infrared spectroscopy, atomic force microscopy, and goniometry. Their biocompatibility was determined using two epithelial cell lines (human umbilical vein endothelial cells and human intestinal epithelial cells), assessing the metabolic activity, cell density, and morphology. The tested cell lines showed different preferences regarding the culture substrate. No clear correlation was found between viability and individual substrate characteristics, suggesting that complex synergistic effects may play an important role in substrate design. These results show that a systematic approach is required to compare the biocompatibility of simple cell culture substrates as well as more complex applications (e.g., bioinks).

## 1. Introduction

With the advent of tissue engineering and regenerative medicine, tissue-specific decellularized extracellular matrices (dECMs) have emerged as excellent sources for advanced cell culture applications. This can be attributed to their close recapitulation of the biochemical and mechanical properties of the histological microenvironment, which enhances cell viability, maturation, and migration [1,2,3,4,5]. Studies have shown that dECM can also be used as a promising source of (bio)printing inks for 3D bioprinting applications in the production of various tissue types, including adipose, cartilage, muscle, cardiac, and liver tissues [1,6,7,8,9,10,11,12]. However, the use of tissue-specific dECMs also has disadvantages. Procurement depends on sacrificial tissues, which are limited in availability and require ethical considerations. In addition, the structure and composition of the starting material may vary from “batch to batch.” Finding alternative sources of scaffold materials that are inexpensive, widely available, and adaptable in a composition that adequately mimics the bio-physico-chemical properties of the native ECM would therefore be an important step toward constructing complex tissues and organs. Several material candidates have been successfully used for tissue engineering, and much research has been done on the biocompatibility of materials and their influence on cell growth and development [13,14,15]. Most original research has been performed using individual cell–material combinations with composite materials optimized for printability and executed in 3D cell culture setups [13,14,15]. Although a large amount of data has been collected and compared in some excellent reviews [13,14,15], a systematic approach comparing the basic interactions of cells with specific polymers and their properties is currently lacking. Thus, both the correlation and the causal chain between individual substrate characteristics and cellular behavior are poorly understood.

The aim of this study was a preliminary selection process for polymers to reduce the total number of experiments required for scaffold design, which rises exponentially with every additional material. It is clear that respective cell cultures have clear preferences in terms of culture substrate [13,14,15]; however, the underlying reasons seem to be complex and require further research. Thus, it seems reasonable to reduce the number of polymers at the 2D level before adding new experimental parameters (such as stiffness, porosity, concentration, etc.) at the 3D level. Therefore, in this work, polymer thin films were prepared using materials that have been previously tested for biocompatibility (either in vivo or in vitro) and systematically compared under the same experimental conditions. Twelve polymers were selected, mainly natural, carbohydrate, and protein-based materials, which are briefly described below. Poly(lactide-co-glycolide) (PLGA) [16,17], which does not fall into either category but is frequently used for bioengineering applications (e.g., with fibroblasts, mesenchymal stem cells (MSCs), or cancer cells) [11,18,19,20], was also examined for comparison. Due to its reversible cross-linking mechanism with divalent cations, alginic acid (ALG) [21,22] has become a very popular polymer for biofabrication and is used in various innovative scaffolding strategies [23,24,25,26,27]. It has found use with several cell-culture applications, including stem cells, chondrocytes, myoblasts, pancreatic islets, etc. [13]. Adding functional groups can also alter cell adhesion and development on ALG substrates [14,27,28]. In this work, sulfated alginate (ALG-S) [21,22,28] was used in addition to its conventional counterpart. Cellulose-based scaffolds are often based on decellularized plant material. Such scaffolds have been successfully repopulated with MSCs, endothelial cells, chondrocytes, HeLa cells, etc. [29,30,31]. Cellulose-derived polymers, such as carboxymethylcellulose (CMC) [32], have also been used in the development of bioinks for applications with keratinocytes, fibroblasts, and endothelial cells [22,33,34,35]. Pullulan (PUL) [36] is a fungus-derived, biodegradable, and non-immunogenic polysaccharide that is FDA approved for many food and pharmaceutical applications [15,37]. Studies have examined its biocompatibility with various cell types, including MSCs, fibroblasts, endothelial cells, and smooth muscle cells [15]. Pullulan has also been successfully combined with dextran in the production of nanoparticles [38]. The biocompatibility of dextran (DEX) [36,39] has already been studied in vitro on cardiomyocytes, fibroblasts, cancer cells, and in vivo [40,41]. In addition, its derivatives have been shown to have antimicrobial effects on Gram-positive and Gram-negative bacteria and yeasts [42]. Hyaluronic acid (HYA) [43,44] is a highly hydrophilic polysaccharide commonly found in animal connective tissue [14,45]. It has been shown to have good biocompatibility with various cell types, including chondrocytes, fibroblasts, endothelial and embryonic cells, and adult stem cells [15,45,46]. Due to its antimicrobial properties, chitosan (CHI) [47,48,49] derived from chitin has also found widespread use in various biomedical applications, including wound dressings and tissue engineering applications, especially for cartilage or bone regeneration [48,50,51]. As one of the main components in animal ECM, collagen (COL) [52,53,54] is an obvious material for tissue engineering and regenerative applications [24,55,56]. Both individually and in combination with other polymers, collagen has been shown to support the growth and development of several cell types, including stem cells, fibroblasts, endothelial cells, astroglia, chondrocytes, etc. [18,57,58] As its derivative, gelatin (GEL) [54,59,60] has also been extensively tested in bioengineering [61]. Its support of successful cell growth has been demonstrated for chondrocytes, fibroblasts, hepatocytes, and human embryonic kidney (HEK) cells [15,62,63]. Using gelatin scaffolds with seeded follicles, researchers at Shah’s laboratory were able to restore fertility in sterile mice [64]. Another protein frequently used in scaffolding is fibrin (FIB-CL) [65], which is naturally produced during blood clotting by an enzymatic polymerization process of fibrinogen (FIB) and thrombin [66,67]. It has found applications in the cultivation of endothelial cells, fibroblasts, stem cells, and neural constructs [13,46]. The final polymer studied in this work was silk fibroin (SILK)[68], which was extracted from silk fibers in a process developed by Rockwood et al. (2011) [69] and has since found applications in biomedical research, including the cultivation of stem cells and fibroblasts [70,71,72].

The surface and chemical characteristics of the thin films were analyzed and compared to the viability and morphological development of two human tissue-derived epithelial cell types, namely, human intestinal epithelial cells (HUIEC), previously isolated and characterized by our group [73], and a line of human umbilical vein endothelial cells (HUVEC). The selected cell types, which form simple (single-layer) epithelia with little ECM, were suitable candidates for cultivation on flat, thin-film coated substrates.

## 2. Results and Discussion

### 2.1. Thin-Film Preparation and Characterization

To minimize the number of variables, all experiments in this study were performed on thin films, which provide an elegant platform for evaluating material properties and cell culture experiments. The thin films were prepared by spin coating polymer solutions (0.5 wt.%) on cleaned, square 1 cm × 1 cm silicon wafers. The samples were dried at ambient conditions and stored at 4 °C for a maximum of 1 week before further use. The structure and composition of the thin films were investigated to verify their successful application and to evaluate correlations between cell viability and material properties of the substrates. Due to varying viscosity, adhesion, and cohesion properties of the solutions, it was expected that the thin films applied by spin coating would have different thicknesses for each material. However, the controlled and repeatable rotation ensured high reliability and reproducibility of the coatings, as previously described [22,74], which was necessary for the qualitative and semi-quantitative comparison of the material properties and cell behavior.

### 2.2. Attenuated Total Reflectance Fourier Transform Infrared Spectroscopy (FTIR-ATR)

ATR-FTIR was used to confirm the presence of the individual polymers on the silicon wafers and to compare the spectra of films to the respective raw materials. Due to a strong background signal of the silicon wafers and the thin coating of the films, the measured spectra showed weaker polymer signals than the raw material and a strong peak between 2500 and 1800 cm^−1^, characteristic of the silicon wafer, which was removed from the final diagram to improve legibility. The corresponding recordings of sample FTIR spectra are shown in Figure 1 (the remaining are in the Appendix A), and the characteristic vibrational frequencies are summarized in Table 1. The signal strengths varied with the materials and likely depended (at least in part) on the thickness of the coating. Protein samples resulted in the strongest signals, which were further observed on PLGA, HYA, and CHI samples. However, even the weakest signals measured (for PUL and DEX) retained some of the characteristic vibrations, confirming the presence of the polymer on the wafer. Of interest is the amide I band measured on the fibrinogen (FIB) and fibrin samples, where a peak shift can be observed (see the Appendix A), which is a characteristic phenomenon in fibrin polymerization [65]. Overall, all coatings showed characteristic vibrations, as described in previous studies, with the individual material references listed in Table 1. These results confirm the successful spin coating of all tested solutions, and all thin layers were further processed and characterized.

### 2.3. Atomic Force Microscopy (AFM)

In order to characterize the surface topology, each thin film was scanned using atomic force microscopy (AFM), which provided roughness parameters and insights into the polymer structure. The results are shown in Figure 2. All recorded samples showed a significant roughness deviation compared to empty silicon wafers (0.1 nm range), which confirms successful coating. In addition to the thin-film roughness, the thickness also appeared to vary. For example, the samples from DEX and PUL showed topographical features visible in the blank samples, whereas they were completely masked in the other samples. Thinner coatings from DEX and PUL were also confirmed by FTIR analysis, as described above. Most of the measured samples showed average roughness values between 0.1 and 1 nm. Thin coatings from COL, HYA, and PLGA showed the highest roughness on the 10 × 10 μm scale (in ascending order). The images of COL, HYA, and PLGA showed the formation of hierarchical structures with higher Sq and Sa values on the 10 × 10 μm scale, whereas the roughness on the 1 × 1 μm scale remained relatively low. This was particularly evident for PLGA, which showed the formation of dispersed particles with a diameter of 0.5–2 μm and an average height of 160 nm (Sq) or 137 nm (Sa). However, individual particles showed a smooth surface with average Sq and Sa values below 0.2 nm.

### 2.4. Contact Angle Measurement

The hydrophilicity of thin films was evaluated by measuring static contact angles with ultrapure water, and the results are shown in Figure 3. All coated samples showed absorption of water into the thin film, resulting in gradual flattening of the applied droplets. Therefore, the values shown here reflect the initial contact angles that appeared immediately after application. After treatment with piranha solution, the blank silicon wafers became significantly more hydrophilic [75], as can be seen from the control samples’ contact angle measurements (45.9°), which was beneficial for coating distribution. With the exception of HYA and ALG, which further increased the hydrophilicity of base substrates, all other samples showed higher contact angles. However, it is noteworthy that the HYA samples may also have contained salt residues formed during thin-film preparation. Therefore, it was expected that the values of pure HYA be higher. Most thin films had a hydrophilic surface with contact angles between 50 and 70°. The only hydrophobic surface was that of fibrinogen at an average of 105.5°, which was significantly higher than expected [76] and was likely the result of the coating process and resulting roughness, which significantly impacts surface wettability [77].

### 2.5. Cell Culture and Viability Analysis

First, the thin films were soaked in the culture medium for 24 h, which was then added to the cell culture in three dilution sets (1:1, 1:2, and 1:4). In addition, the cells were cultured directly on the thin-film surface. Upon contact with the prepared thin films and their extracts, cell viability was assessed by measuring the metabolic activity of the cells using the MTT assay and examining cell morphology by fluorescence staining and subsequent microscopy. In the described experimental design, two main ways in which the polymers could affect the cells must be considered:Chemical interactions: The polymers may facilitate or prevent cell attachment, trigger or block membrane receptors, and act as chelators for nutrients or toxic compounds. For example, CHI is generally considered biodegradable and biocompatible with simultaneous antimicrobial activity, typically attributed to free amino groups [51,78].Structural interactions: Surface roughness and solubility of the polymers determine the attachment surface and its stability and change in viscosity of the culture medium near the thin-film surface.Depending on the solubility of the thin films in the nutrient medium under cell culture conditions (37 °C, 5% CO_2_), the cell-polymer interaction may vary. For soluble thin films, the effect on cell viability should be comparable for direct contact and exposure to extracts, with a correlation between metabolic activity and polymer concentration. For poorly soluble polymers, thin-film extracts are expected to have less effect on cell viability. It is important to note that surface properties such as roughness can only affect the cells if the polymer is poorly soluble and therefore sufficiently stable. In addition to the thin-film properties measured in this study, the available literature was analyzed to compare the influence of thin film on cell development with functional groups of the polymers with which they might interact. Specifically, the polymers were analyzed for hydroxy, carboxyl, or amino groups. The polysaccharides selected for this study consist of repeating hexoses with respective functional groups so that their number per molecular weight could be easily derived from the chemical structure of the individual polymers, as found in the literature [27,28,35,37,42,45,51]. The same method of assessing the functional groups per unit molecular weight was also used for PLGA [20]. With their complex amino acid sequences and folding and intertwining chains, the free functional groups of proteins are more difficult to estimate. Therefore, in the work presented here, the number of functional groups was estimated from the proportion of individual amino acids and their functional groups in the total composition alone. E.g., for silk fibroin, which is produced by the silk moth (*Bombyx mori*), the composition was estimated according to the genetic sequence, coding for 45.9% glycine, 30.3% alanine, 12.1% serine, 5.3% tyrosine, and 1.8% valine [72]. The same procedure was used to determine the amount of functional groups in COL [56], GEL [61], and FIB [66]. For the reasons mentioned above, the number of functional target groups per 100 kDa in proteins is only a rough estimate. It does not necessarily represent the actual value of the functional groups available for interaction with cultured cells.

#### 2.5.1. HUVEC

The results of the biocompatibility analysis of prepared thin films and HUVEC cells are shown in Figure 4 (with additional information in the Appendix A). The results of the viability analysis (obtained with the MTT test and fluorescence microscopy, shown on the left Y-axis) are plotted against the thin-film characterization data (shown on the right Y-axis) as described in Section 2. Results and Discussion. Most tested materials showed good biocompatibility with the cells. The highest relative viability values at attachment were measured on PLGA substrates with almost double the metabolic activity and cell count compared to the bare Si wafer. High values were also observed on collagen, fibrin, hyaluronic acid, and gelatin substrates in descending order. The lowest viability was again observed when the cells were cultivated on chitosan substrates, and poor biocompatibility with HUVECs can also be attributed to alginate. The remaining thin films showed the growth and viability of cells similar to the control sample.

#### 2.5.2. HUIEC

Cells cultured on collagen showed the highest metabolic activity, cell density, and cell size of all samples (data in the Appendix A). The latter was also the only substrate that seemed to be more effective for culturing than the control sample. The other materials resulted in lower viability than the Si wafers, but to a relatively small extent. Viability was in the range of 80–100%, which may be suitable for certain cell culture applications. The only sample that prevented normal cell development was chitosan, with very few cells observed, which also retained a small and circular shape as a sign of poor adhesion. An estimated biocompatibility score (described further in Section 3. Materials and Methods) was attributed to the materials for both cell lines and compared to the thin-film characteristics (see Figure 5). The purpose of the score was to improve the legibility of the data and allow an evaluation of how individual properties impact viability. A non-parametric Spearman’s test was performed to statistically analyze possible correlations between thin-film characteristics and individual viability values (MTT results for attachment, extracts, cell number, and cell sizes, respectively).

### 2.6. Interpretation of the Results

COL seemed to be a very suitable material for both cell lines, which is not surprising given its general presence in extracellular matrices and in line with previous studies that showed successful cultivation of vascular and intestinal epithelium on collagen-based substrates [79,80,81]. For vascular endothelial cells, the highest biocompatibility was attributed to PLGA, which has also been used in vascular tissue engineering and stent fabrication [20,82]. The material that proved unsuitable for both cell lines was CHI. Although it had already been used in bioengineering and was considered biocompatible, CHI nanoparticles also proved to be cytotoxic to liver cells [83]. Surprisingly, the MTT results using CHI extracts showed a positive effect on cell viability. The material is poorly soluble at neutral and high pH, suggesting that either very small amounts of CHI could be beneficial or that this polymer chelates compounds that would otherwise slow down cells’ metabolic activity. Most other materials seemed to be suitable as culture substrates for both cell types. A more extensive quantitative study will be necessary to show how a change in the amount of material available for interaction with the cell culture will affect growth and development. However, the polymer concentrations used in this study to produce thin films were already lower than those typically used in hydrogels for scaffolding and tissue engineering.

Although both cell lines tested showed clear and specific preferences regarding the growth substrate, no direct correlation between biocompatibility and individual material properties that might influence the interaction could be established. This suggests that synergistic effects between substrate properties are likely to play an important role in the development of tissue engineering scaffolds or advanced wound-healing materials. Furthermore, since the materials tend to change their structural properties when immersed from a dry environment at room temperature into a culture medium at 37 °C, it is difficult to predict suitable substrates based on the available data alone. Variable cell-type preferences for specific growth substrates refute the possibility of developing universal “bioinks” with optimal biocompatibility for 3D bioprinting and tissue engineering [84,85]. On the other hand, this presents an opportunity to control the development of individual cell lines within co-cultures or 3D environments that are closer in structure and biochemical cues to their native tissue environment. The empirical determination of cell growth and viability also seems feasible when using the experimental setup described above. It suggests developing a “biocompatibility roadmap” for cell-substrate combinations that will provide researchers and engineers with a tool for material selection and bioink design. In addition, the improvement of cultivation substrates could significantly reduce the time and costs for general cell culture work and, with the advent of cellular agriculture, the agri-food sector as well.

For the successful development of a biocompatibility roadmap, several unanswered questions need to be addressed. These include the characterization of thin films under cell culture conditions and, in particular, the cell–material interface with a focus on binding sites, substrate stability, and possible substrate modifications that allow stable attachment while permitting cell mobility. To better understand cell–material interactions, a more quantitative approach to culturing will likely be required, experimenting with different cell-seeding densities and over longer culturing periods. A live cell-imaging system that provides continuous insight into the culture during incubation would greatly benefit this effort while reducing the number of parallel experiments required for individual time points. Furthermore, additional experiments should be performed on thicker polymer substrates to significantly increase the influence of substrate rheology on the overall interaction. Such experiments can be performed both as a flat monolayer (2D) on thick polymer deposits or their hydrogels and encapsulated in the material (3D).

## 3. Materials and Methods

### 3.1. Thin-Film Preparation

As the substrate for the thin films, silicon wafers (Topsil, Frederikssund, Denmark) were used, which were cut into square 1 cm^2^ pieces and cleaned thoroughly, as validated previously [33,86]. Each piece was rinsed with 70% ethanol and subsequently with ultrapure (18.2 mΩ cm at 25 °C) water. After rinsing, each silicon sample was soaked in piranha solution (H_2_O_2_ (30%) and H_2_SO_4_ (concentrated) combined in a mixture of 1:7 *v/v*) for 15 min at room temperature, followed by rinsing with and soaking in ultra-pure water for an additional 15 min. Finally, each sample was dried in a stream of dry, high purity N2 (99.99%, Messer, Maribor, Slovenia). A total of 0.5% aqueous solution of the selected natural polymers was prepared according to manufacturer specifications, with certain described exceptions, summarized in Table 2. All chemicals were obtained from Sigma-Aldrich unless otherwise specified. To prepare the polymer thin films, the solutions were spin coated onto cleaned silicon wafers. A layer of polymer solution was deposited by carefully applying 50 μL on a fixated wafer and rotating the sample at 2500 RPM for 180 s using the SPIN-150i-NPP spin coater (SPS Vertriebs GmbH, Berlin, Germany). The process was repeated three times. Afterward, the samples were stored at 4 °C before further use. Thin films for gelatin and fibrinogen were prepared in duplicate, with one series undergoing cross-linking to produce more stable coatings. Gelatin cross-linking was performed as described previously [64]. Briefly, the samples were soaked in an aqueous solution of 15 mM *N*-(3-Dimethylaminopropyl)-*N*′-ethylcarbodiimide and 6 mM *N*-Hydroxysuccinimide for 30 min at room temperature, followed by careful rinsing using ultrapure dH_2_O and air drying at room temperature overnight. To cross-link fibrinogen and form fibrin, the silicon wafers with thin films were soaked in a 20 mM CaCl_2_ solution with 1 U/mL of thrombin for 30 min at room temperature. Subsequently, the samples underwent careful rinsing using ultrapure water.

### 3.2. Attenuated Total Reflectance (ATR) FTIR

To confirm polymer presence on the wafers and their chemical characteristics, Fourier-transform infrared (FTIR) spectra were recorded by measuring the attenuated total reflection (ATR) using a Cary 630 FTIR spectrometer (Agilent, Santa Clara, CA, USA). A spectral range between 4000 and 650 cm^−1^ (with a resolution of 2 cm^−1^) was recorded for each sample, as described previously [48,86,87,88,89].

### 3.3. Atomic Force Microscopy (AFM)

Atomic force microscopy (AFM) was used to evaluate surface topology and surface roughness parameters of the samples using a Keysight 7500 AFM (Keysight Technologies, Santa Rosa, ON, Canada) as described before [33,48,86,88,89]. Images were recorded in tapping mode with silicon-based tips (ATEC-NC-20, Nanosensors, Neuchatel, Switzerland). Each sample was scanned at room temperature using a resonance frequency of 210–490 kHz and a force constant of 12–110 N/m. Images in 1024 × 1024 pixel resolution were recorded on areas of 10 × 10 and 1 × 1 μm^2^. The surface was scanned at 3 locations on the sample to verify the obtained data’s consistency. The images were further processed using Gwyddion 2.55 software to calculate the roughness values according to ISO 25178, namely, the root mean square height (Sq) and the arithmetical mean height (Sa) values and visualization of the sample topographies.

### 3.4. Contact Angle Measurements

To determine the samples’ hydrophilicity, the static contact angles were measured using the sessile drop technique, whereby ultrapure water was applied to the thin films using a custom-made goniometric setup. Five 1 µL droplets per thin film were applied and measured at room temperature and 30% relative humidity. The entire application process was recorded on video, and the images selected for contact angle calculation were taken immediately after drop deposition. The static contact angles were then evaluated by measuring the enveloping rectangle with ImageJ and the height-width method.

### 3.5. Cell Selection and Culturing

Two lines of epithelial cells were examined in this work, namely, human intestinal epithelial cells (HUIEC) that were previously isolated and characterized by our group [73] and a line of human umbilical vein endothelial cells (HUVEC) purchased from ATTC. A base culture of cells was maintained in Advanced Dulbecco’s Modified Eagle’s Medium (ADMEM, Gibco, Thermo Scientific, Waltham, MA, USA), supplemented with 5% (wt) fetal bovine serum (FBS, Gibco, Thermo Scientific, Waltham, MA, USA) at 37 °C and 5% CO_2_.

### 3.6. MTT (3-(4,5-Dimethylthiazol-2-yl)-2,5-diphenyltetrazolium Bromide) Test

Cell viability was determined using the MTT assay 24 h after contact with the polymers. Metabolically active cells reduce the yellow-colored 3-[4,5-dimethylthiazol-2-yl]-2,5 diphenyltetrazolium bromide (MTT) into formazan. The latter forms purple crystals, soluble in organic solvents such as dimethyl sulfoxide (DMSO). Since formazan’s photometric absorption (at 570 nm) shows a linear correlation with the overall cell viability, the assay allows for accurate quantification [90,91,92]. The assay was performed as described previously [48,86,89], with specific details below.

### 3.7. Cell Culture with Thin-Film Extracts

In the first part of the biocompatibility assessment, the extract or elusion test was carried out [48,86,89]. The thin films coated with biomaterials were sterilized under UVC irradiation for 30 min and soaked in 1.5 mL of ADMEM supplemented with 5%.(wt) FBS and incubated for 24 h at 37 °C and 5% CO_2_. In parallel, HUVEC and HUIEC cells were seeded into P96 microtiter plates at a concentration of 10,000 cells/well and allowed to attach for 24 h. Next, the extracts’ serial dilutions were prepared in the ratios of 1:1, 1:2, and 1:4 and transferred onto the cell layer, followed by incubation at 37 °C and 5 wt.% CO2 for 24 h. Each combination was carried out in four replicates, and additional control samples were prepared, with cells incubated in ADMEM supplemented with 5 wt.% FBS. Following incubation, cell viability was determined using the MTT assay as described above. The medium was removed, and the MTT reagent (10 μL MTT + 90 μL medium) was added to the wells and incubated for 2–4 h. After incubation, the MTT reagent was removed, and 100 μL of DMSO was added to each well. After 5 min, the crystals had dissolved, and the absorbance was measured at 570 nm using a Varioskan Flash multi-well plate reader (Thermo Scientific, Waltham, MA, USA).

### 3.8. Cell Culture on Thin Films

Cell viability was also determined when in direct contact with the thin films. The silicon wafer-thin film samples were placed in P24 microtiter plates and sterilized by UVC irradiation for 30 min. Following sterilization, HUVEC and HUIEC cells were seeded on the thin films at a concentration of 50,000 cells/well in 1 mL of ADMEM supplemented with 5% FBS. Each experimental condition was tested in three replicates, and a non-coated silicon wafer was used as a control. The samples were incubated for 48 h at 37 °C and 5% CO_2_, followed by a viability analysis using an MTT test. After incubation, the silicon wafer plates with cells were transferred to a new P24 well plate containing MTT reagent (50 μL MTT + 450 μL medium). After 2–4 h the MTT reagent was removed, and crystals were dissolved in 250 μL of DMSO. A total of 200 μL of each sample was then pipetted onto a P96 plate, and the absorbance was measured at 570 nm using a Varioskan Flash multi-well plate reader (Thermo Scientific, Waltham, MA, USA).

### 3.9. Cell Morphology Analysis

To assess the cell adhesion and morphology of the cells growing on the thin films, phalloidin and DAPI staining were used to visualize the actin filaments and cell nucleus, respectively. The same experimental conditions were applied as for the cell viability testing. After 48 h of incubation on the thin films, the medium was removed and the cells were incubated for 15 min at room temperature with a fixing solution (Sigma-Aldrich). Fixation was followed by washing with PBS (3 × 15 min) and subsequent staining using CytoPainter Phalloidin-iFlour 555 (1:1000 in PBS with 1% BSA) for 90 min at room temperature in total darkness. Before imaging, the staining solution was removed and the samples were covered with a drop of mounting medium (Fluoroshield containing DAPI, Sigma-Aldrich, Burlington, VT, USA) and incubated for 5 min in total darkness. Imaging was performed using the EVOS cell-imaging system.

### 3.10. Statistical Analysis

Statistical analysis was performed using SPPS Statistics 25 (IBM Corp., Armonk, NY, USA). Normality distribution was determined using the Kolmogorov–Smirnov test and range of skewness and kurtosis. None of the dependent variables were normally distributed and are therefore presented as median and 95% confidence interval. For statistical analysis, the non-parametric Spearman’s test was chosen to test the correlation between dependent and independent variables. *p*-values < 0.05 were considered statistically significant. All statistical data is available in the Appendix A

### 3.11. Biocompatibility Score

For all MTT test results, as well as cell size and cell number (evaluated using ImageJ), the relative values were calculated, and thus the control sample, taking a value of 1 (100%). The obtained values were then corrected with respective weighing factors to account for the differences in environmental impact on the cells. Finally, the total score was determined by the sum of all weighed values, according to Equation (1):(1)S=4a+b+0.5c+0.25d+2e+2f. 

Equation (1): Calculating the biocompatibility score (***S***), where ***a*** equals MTT results of cells grown on thin films, ***b*** equals MTT results of 1:1 extracts, ***c*** equals 1:2 extracts, ***d*** equals 1:4 extracts, ***e*** equals the relative number of cells observed in the morphological analysis, and ***f*** equals the relative cell size.

## 4. Conclusions

This study highlights the specific interaction between vascular and intestinal epithelial cells with different substrates. The results show that the respective cells have clear preferences in regard to growth substrate; however, the growth and development of one cell line do not necessarily provide insight into another cell line’s potential behavior on the same set of substrates.

Therefore, the need for a systematic approach to substrate selection for both basic and advanced cell culture applications is becoming increasingly clear. In the transition from traditional to three-dimensional cell culture, this aspect will be crucial in designing and developing tissue-engineering scaffolds, wound-healing materials, drug-delivery systems, and other biomedical applications. However, as the set of possible experiments rises exponentially with every subsequent material–property combination, it seems feasible to narrow the substrate selection for testing on flat surfaces before continuing to the third dimension. Building on the established framework, a roadmap for cell–material interaction can be developed, providing guidelines for selecting components in coatings, scaffold design, and other biomedical applications.

## Figures and Tables

**Figure 1 polymers-13-02311-f001:**
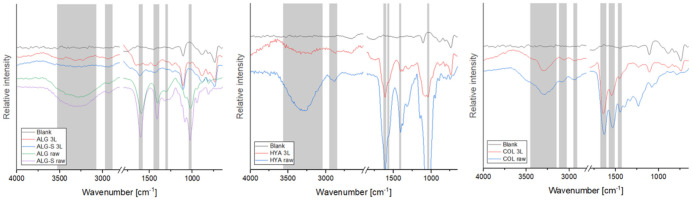
FTIR spectra of ALG, ALG-S, HYA, and COL thin films compared with the spectra of raw polymers and Si wafers. The characteristic bands are marked in grey, with the respective frequencies listed in Table 1 (spectra of the remaining polymers and thin films are available in the Appendix A). All carbohydrate samples show characteristic OH bands at about 3300 cm^−1^, with additional vibrations for CH and CO bonds. In addition, hyaluronic acid and chitosan show NH and CN vibrations, which are characteristic for amino groups. The protein samples show peaks for amide I, II, and III at polymer specific frequencies and additional vibrations for CH and CO groups. PLGA has a strong and narrow peak at 1750 cm^−1^, characteristic for its ketone groups, and additional peaks for CH and C-O groups.

**Figure 2 polymers-13-02311-f002:**
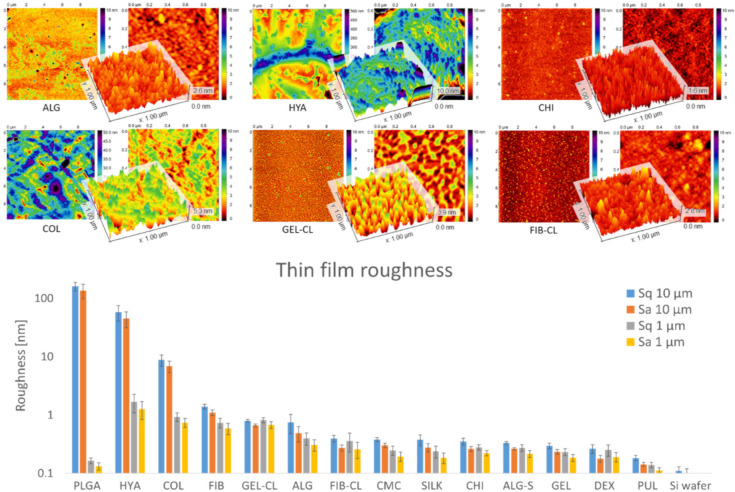
Results of AFM analysis. The calculated roughness values (Sq and Sa of 10 × 10 and 1 × 1 μm scans) of all prepared thin films are shown in the chart (**bottom**), as well as micrographs of ALG, HYA, CHI, COL, GEL-CL, and FIB-CL thin films (**top**—the remaining micrographs are available in the Appendix A). For each sample, areal scans of 10 × 10 μm (left) and 1 × 1 μm (right) are shown, with a 3D render (**middle**). The height variation of most polymers lies within the 0–10 nm range, except for collagen (0–50 nm), and hyaluronic acid and PLGA (in the 0–500 nm range).

**Figure 3 polymers-13-02311-f003:**
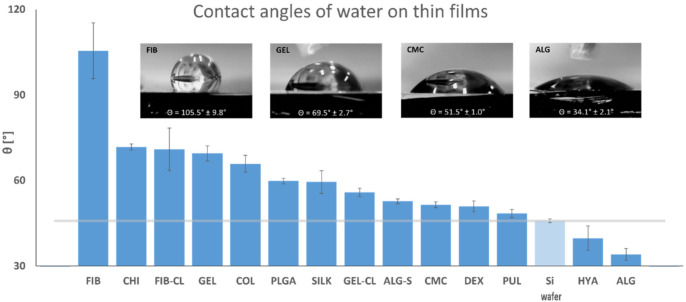
The hydrophilicity of thin films**.** Static contact angles were determined using the sessile drop method, depositing 1 μL droplets of ultrapure water and image capturing after initial contact. The results are summarized in the chart, with the average contact angle of water on clean Si wafers represented by the grey horizontal line. Images of water droplets on FIB, GEL, CMC, and ALG thin films are shown for comparison.

**Figure 4 polymers-13-02311-f004:**
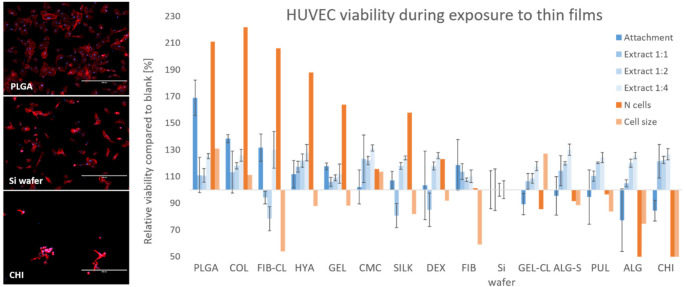
Biocompatibility of the thin films to HUVEC cell culture**.** Images of HUVEC cells grown on thin films of PLGA, CHI, and the control sample are shown, stained with DAPI and Phalloidin (**left**). Scale bars indicate a length of 400 μm. The results of the MTT assay (blue), as well as the evaluation of the morphological analysis (orange), are summarized in the chart (**right**). The bars represent the relative values compared to the control sample (100%). Cells grown on PLGA thin films demonstrated the highest metabolic activity in the MTT assay, as well as high cell density and cell size.

**Figure 5 polymers-13-02311-f005:**
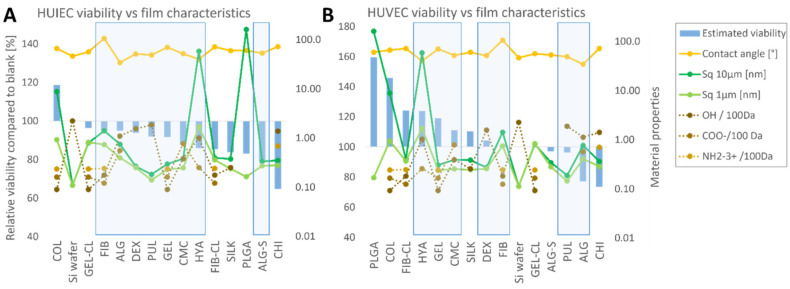
Estimated viability of HUIEC (**A**) and HUVEC (**B**) cells compared to thin-film characteristics. For improved legibility of the data, the values of the estimated viability were calculated by combining the results of the MTT assay and cell morphology into a biocompatibility score as shown in Figure 4 and further described in the Materials and Methods section. The estimated viability values are compared to thin-film characteristics (curves plotted against the secondary Y axis). The static water contact angle (yellow), roughness (green), and the number of functional groups (brown) are shown. Water-soluble polymers are marked with a blue frame.

**Table 1 polymers-13-02311-t001:** Characteristic vibration frequencies of polymer thin films measured using ATR-FTIR and compared to data from previous studies.

Vibration (cm^−1^)	ALG [21,22]	ALG-S [21,22]	CMC [32]	PUL [36]	DEX [36,39]	HYA [43,44]	CHI [47,48,49]
ν(OH)	3300	3300	3300	3330	3220	3300	3300
ν(CH)_anomer_	2930	2930	2930	2920	2900	2925	
ν(COO)_asym_	1595	1595	1595				
ν(COO)_sym_	1425	1425	1415			1411	
δ(CCH)+δ(OCH)	1300	1300	1310				
ν(C-O)	1024	1024					
ν(NH)_sym_						3300	3300
Amide I						1650	1667
						1614	
ν(NH_2_)						1560	1560
ν(C-N)						1310	1309
ν(C-O-C)						1043	
ν(C-O)				1155	1155		
				1107	1107		
**Vibration (cm^−1^)**	**COL** [52,53,54]	**GEL** [54,59,60]	**GEL-CL** [54,59,60]	**FIB** [65]	**FIB-CL** [65]	**SILK** [68]	**PLGA** [16,17]
ν(NH)	3300					3290	
ν(CH)						3060	3000
Amide B band	2928	2940	2940			2030	
ν(C=O)							1750
Amide I	1600–1700						
Amide II	1544	1525	1525	1520	1520	1520	
δ(C-H_2_)	1454	1450	1450	1450	1450	1410	
δ(C-H_3_)	1390	1400	1400	1390	1390	1380	
Amide III	1236	1235	1235	1240	1240	1230	
ν(C–N)				1300	1300		
ω(C-H_2_)						1330	
ν(C-H) methyl							1450
ν(C-O-C)						1160	1085
ν(C–O)	1035						

**Table 2 polymers-13-02311-t002:** A summary of the materials used and prepared solutions.

Polymer	Recipe
ALG	A total of 0.01 g was dissolved in 2 mL of high-purity dH_2_O with agitation on a magnetic stirrer at room temperature until completely dissolved.
ALG-S	A total of 0.01 g of dry material obtained from Zenobi Group (ETH Zürich, Switzerland) [28] was dissolved in 2 mL of high-purity dH_2_O with agitation on a magnetic stirrer at room temperature until completely dissolved.
CMC	A total of 0.01 g was dissolved in 2 mL of high-purity dH_2_O with agitation on a magnetic stirrer at room temperature until completely dissolved.
PUL	A total of 0.01 g was dissolved in 2 mL of high-purity dH_2_O with agitation on a magnetic stirrer at room temperature until completely dissolved.
DEX	A total of 0.01 g was dissolved in 2 mL of high-purity dH_2_O with agitation on a magnetic stirrer at room temperature until completely dissolved.
HYA	A total of 0.01 g was dissolved in 2 mL of phosphate buffered saline (PBS) with agitation on a magnetic stirrer at 90–95 °C until completely dissolved, followed by cooling to 37 °C.
CHI	A total of 0.01 g was dissolved in 2 mL of 17 mM solution of acetic acid with agitation on a magnetic stirrer at room temperature until completely dissolved.
GEL	A total of 0.01 g was dissolved in 2 mL of high-purity dH_2_O with agitation on a magnetic stirrer at 40 °C until completely dissolved and cooled to room temperature.
COL	A total of 0.01 g was dissolved in 2 mL of 0.2 M acetic acid with agitation on a magnetic stirrer at 45 °C overnight.
FIB	A total of 0.01 g was dissolved in 2 mL of 0.9% NaCl with agitation on a magnetic stirrer at room temperature until completely dissolved.
SILK	A prepared solution was obtained from the department of nanostructured materials (IJS, Ljubljana, Slovenia) and prepared as previously described [69,70].
PLGA	A total of 0.01 g was dissolved in 2 mL of acetone with gentle manual agitation until completely dissolved.

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
