# Peer review of "Investigating the Viability of Epithelial Cells on Polymer Based Thin-Films"

_polymers, 2021, doi:10.3390/polym13142311_

Round 1

Reviewer 1 Report

In the study, polymer thin films previously tested for biocompatibility (either in vivo or in vitro) are prepared and systematically compared under the same experimental conditions. Twelve polymers were selected, mainly natural, carbohydrate and protein based materials. Another material Poly(lactide-co-glycolide) (PLGA), frequently used for bioengineering applications (e.g., with fibroblasts, mesenchymal stem cells (MSCs), or cancer cells), is also examined for comparison.

Surface and chemical characteristics of the thin films are analysed and compared to the viability and morphological development of two human tissue-derived epithelial cell types (intestinal epithelial cells (HUIEC), and a line of human umbilical vein endothelial cells (HUVEC).

The method of characterization includes total reflectance Fourier transform infrared spectroscopy, AFM, Contact angle measurement, Cell culture and viability analysis.

Most of the tested materials showed a good biocompatibility with the two cell types.

The results show that no direct correlation between biocompatibility and individual material properties that might influence the interaction can be established.

- This result is true in the limit of surface roughness. Beyond a surface roughness threshold, the interaction of cells and the surface is amplified. This effect would be important to underline.

According to the presented results the goal of the study is to present a roadmap for cell-material interactions, providing guidelines for selecting components in coatings, scaffold design, as well as other biomedical applications.

The study is well documented and all the necessary studies to confirm the biocompatibility are presented.

However, I regret that the part (interpretation of the results) is short. I can understand that a screening of the materials for the best adapted materials for biocompatibility is necessary but no additional test is proposed to discriminate among the different materials. Moreover, other questions concerning the time for cell growing, the directionality of the cell growing, the possibility to functionalize the surface of the materials and the possibility to change the surface, are not addressed even briefly as a possible extension of the work. 

Author Response

We thank the reviewer for the comments and constructive criticism. We have expanded the interpretation/discussion section to include the raised issues in the revised manuscript, which now includes the following text:

For the successful development of a biocompatibility roadmap, several unanswered questions need to be addressed. These include the characterization of thin films under cell culture conditions and, in particular, the cell-material interface with a focus on binding sites, substrate stability, and possible substrate modifications that allow stable attachment while permitting cell mobility. To better understand cell-material interactions, a more quantitative approach to culturing will likely be required, experimenting with different cell seeding densities and over longer culturing periods. A live cell imaging system that provides continuous insight into the culture during incubation would greatly benefit this effort while reducing the number of parallel experiments required for individual time points. Furthermore, additional experiments should be performed on thicker polymer substrates to significantly increase the influence of substrate rheology on the overall inter-action. Such experiments can be performed both as a flat monolayer (2D) on thick polymer deposits or their hydrogels and encapsulated in the material (3D).

Reviewer 2 Report

Development of tissue engineering and regenerative medicine is a challenging task of modern interdisciplinary science. To achieve an ultimate goals a deep insight into the processes at atomic and molecular level scale is required. In this regard the present work evaluating the viability of vascular and intestinal epithelial cells on different polymers by a combination of complimentary structural and chemical studies is quite important. One of the interesting findings is that, via investigating the possible correlations between increased cell viability and material properties, the tested cell lines show different preferences regarding the culture substrate, and that no clear correlation between viability and individual substrate characteristics is found,
The strength of the work: Combination of the state-of-the art sample preparation and complimentary characterization techniques, enabling quite unambiguous data collection. High level of data interpretation and analysis.
The weakness: The work would gain if there were evidence of the character of molecular interaction of the cell and the substrate at the very interface, the interaction that presumably governs the film overall morphology.  
In general the work is of quite high level, its design is logical and clear, quality of presentation is high, figures are clear and informative, reference list is comprehensive, quite complete and up-to-dated. In my view, the manuscript is suitable for publication in Polymers in its present form.

Author Response

We thank the reviewer for their analysis and comments. We agree that investigation of the cell-substrate interface would increase the quality, however, this was beyond the scope of the presented work. Therefore we added a brief description to the discussion/outlook section, which now includes the following text:

For the successful development of a biocompatibility roadmap, several unanswered questions need to be addressed. These include the characterization of thin films under cell culture conditions and, in particular, the cell-material interface with a focus on binding sites, substrate stability, and possible substrate modifications that allow stable attachment while permitting cell mobility. To better understand cell-material interactions, a more quantitative approach to culturing will likely be required, experimenting with different cell seeding densities and over longer culturing periods. A live cell imaging system that provides continuous insight into the culture during incubation would greatly benefit this effort while reducing the number of parallel experiments required for individual time points. Furthermore, additional experiments should be performed on thicker polymer substrates to significantly increase the influence of substrate rheology on the overall inter-action. Such experiments can be performed both as a flat monolayer (2D) on thick polymer deposits or their hydrogels and encapsulated in the material (3D).